# Differential Polarization Imaging of Plant Cells. Mapping the Anisotropy of Cell Walls and Chloroplasts

**DOI:** 10.3390/ijms22147661

**Published:** 2021-07-17

**Authors:** Jasna Simonović Radosavljević, Aleksandra Lj. Mitrović, Ksenija Radotić, László Zimányi, Győző Garab, Gábor Steinbach

**Affiliations:** 1Department of Life Sciences, Institute for Multidisciplinary Research, University of Belgrade, Kneza Višeslava 1, 11000 Belgrade, Serbia; jasna@imsi.rs (J.S.R.); mita@imsi.rs (A.L.M.); xenia@imsi.rs (K.R.); 2Institute of Biophysics, Biological Research Centre, Eötvös Loránd Research Network, 6726 Szeged, Hungary; zimanyi.laszlo@brc.hu; 3Institute of Plant Biology, Biological Research Centre, Eötvös Loránd Research Network, 6726 Szeged, Hungary; garab.gyozo@brc.hu; 4Department of Physics, Faculty of Science, Ostrava University, 709 00 Ostrava, Czech Republic; 5Biofotonika Research and Development Ltd., 6720 Szeged, Hungary; 6Cellular Imaging Laboratory, Biological Research Centre, Eötvös Loránd Research Network, 6726 Szeged, Hungary

**Keywords:** anisotropy, cell wall, chloroplast, circular dichroism, DP-LSM, fluorescence-detected linear dichroism, micro-spectropolarimetry, molecular organization, polarized light, RCM

## Abstract

Modern light microscopy imaging techniques have substantially advanced our knowledge about the ultrastructure of plant cells and their organelles. Laser-scanning microscopy and digital light microscopy imaging techniques, in general—in addition to their high sensitivity, fast data acquisition, and great versatility of 2D–4D image analyses—also opened the technical possibilities to combine microscopy imaging with spectroscopic measurements. In this review, we focus our attention on differential polarization (DP) imaging techniques and on their applications on plant cell walls and chloroplasts, and show how these techniques provided unique and quantitative information on the anisotropic molecular organization of plant cell constituents: (i) We briefly describe how laser-scanning microscopes (LSMs) and the enhanced-resolution Re-scan Confocal Microscope (RCM of Confocal.nl Ltd. Amsterdam, Netherlands) can be equipped with DP attachments—making them capable of measuring different polarization spectroscopy parameters, parallel with the ‘conventional’ intensity imaging. (ii) We show examples of different faces of the strong anisotropic molecular organization of chloroplast thylakoid membranes. (iii) We illustrate the use of DP imaging of cell walls from a variety of wood samples and demonstrate the use of quantitative analysis. (iv) Finally, we outline the perspectives of further technical developments of micro-spectropolarimetry imaging and its use in plant cell studies.

## 1. Introduction

To understand the functions of any biological structure, it is essential to acquire detailed and accurate information about its architecture at different length scales. Light microscopy imaging techniques are of outstanding position for several reasons: (i) Light microscopes cover the range of several hundred nanometres to millimetres, with wavelength-dependent spatial resolution limits of about 200 nm in the lateral direction and about 500 nm in the axial direction [1,2,3,4]. (ii) This resolution limit can be improved by a factor of 1.4 using the procedure of rescanning (RCM) [5], or, more substantially, by applying super-resolution technologies [6]. (iii) Single molecules or molecular assemblies—in a ‘dark’ background—can be made visible and tracked, even if they are much smaller than the diffraction limit of the microscopy. This is evident for a fluorescent molecule in a nonfluorescent background, but other distinct features, such as a single molecule with strong Raman signal or a (set of) well-aligned transition dipole(s) with strong anisotropy in an isotropic matrix, can emerge as distinct data points in the image. Of course, in all these cases, the microscope must be made capable of measuring the corresponding quantity, which—after proper scaling—can be displayed as a complementary image and overlaid on the ‘conventional’ transmission or confocal fluorescence image. (iv) There are numerous possibilities to supplement the ‘conventional’ microscopy images with complementary images, for example, FLIM (the fluorescence lifetime imaging microscopy) [7], TIRF (the total internal reflection fluorescence microscopy) [8], and confocal Raman imaging microscopy [9]. However, some of these measuring techniques cannot readily be combined with the ‘conventional’ laser-scanning excitation and the confocal detection of the fluorescence of the sample. We have shown earlier that differential polarization (DP) measurements can be relatively easily conjoined with the ‘conventional’ imaging regimes of LSM [10,11] and RCM [12]. This can be performed by using the theory and technologies of spectropolarimetry and DP measuring procedures [13,14].

In the following sections: (i) to justify the use of DP imaging, we outline the wealth of information which can be obtained by studying interactions of highly organized molecular assemblies with polarized light, and briefly describe the theories and backgrounds of DP imaging; (ii) we give a short overview of the advancement of DP microscopy, from scanning stage to laser-scanning and rescan techniques and their different configurations; and (iii) show the use of DP-LSM and DP-RCM techniques to map the anisotropic molecular organization of chloroplast thylakoid membranes and plant cell walls; (iv) finally, we give considerations of potential further applications of DP microscopy in revealing the self-assembly, molecular organization, and structural dynamics of plant cells and their constituents.

## 2. Differential Polarization Imaging Theory and Devices

### 2.1. Advantages and Need of Spectropolarimetry Techniques in LSMs

To substantiate the advantages of using controlled or modulated polarized light instead of nonpolarized light, both when exciting the sample with a laser beam and when analysing the transmitted, reflected, or fluorescence emission light in a microscope, the following considerations can be given.

#### 2.1.1. Light-Matter Interactions—Mueller Matrix

The polarization state of light can conveniently be characterized by the so-called Stokes parameters, often given as a vector (*I*, *Q*, *U*, *V*). For a monochromatic light beam propagating along the z-axis, these vector components, respectively, describe the intensity, linear polarization content (in the xz (0°) vs. yz (90°) directions and +45° vs. −45°), and the amount of right- or left-handed circular polarization of the beam. When a beam with initial states (*I*_0_, *Q*_0_, *U*_0_, *V*_0_) interacts with a sample (which absorbs or scatters the beam), its polarization content will almost always be changed, and, thus, the emerging beam will be characterized by a different Stokes vector. The interaction is characterized by a 4 × 4 matrix, the Mueller matrix [15], the elements of which, from M_11_ to M_44_, carry physical information regarding the molecular organization of the sample. In transmission mode, only one (the absorbance, M_11_) of the 16 elements operates with nonpolarized light; the determination of all other elements requires the use of linearly or circularly polarized light. In complex materials, all elements are independent and carry unique physical information about the molecular organization of the sample [16]. To obtain at least partial information about the anisotropic organization of the sample, linear dichroism (LD, M_12_ and M_13_) and circular dichroism (CD, M_14_) measurements are performed, which, respectively, provide information about the alignment of the absorption dipoles with respect to a predefined co-ordinate system and the asymmetry (chirality) of molecules or of chiral interactions in molecular assemblies. Here, for instance for CD, M_14_ indicates that alternatively left and right circularly polarized light beams are incident on the sample, and the intensity of the transmitted light (the absorbance) is measured without investigating its polarization content and, thus, with the indices, referring to the 4th and 1st components of the Stokes vector. It is interesting to note that M_41_, i.e., measuring the circular polarization content after the interaction of a nonpolarized measuring beam, at least on green leaves, yielded very similar spectra to the conventionally measured M_14_ (CD) spectra [17].

#### 2.1.2. Polarization State of the Fluorescence

Regarding the fluorescence, the physical quantities analogous to LD and CD are the content of the linear and circular polarization of the emitted light (r and CPL, respectively). Evidently, the polarization state of the fluorescence light, upon nonpolarized excitation, is determined by the molecular organization parameters controlling the fluorescence emission. In addition to r and CPL, P, the degree of polarization of the fluorescence emission, elicited by polarized excitation, is a frequently determined quantity in spectropolarimetry. (For definitions and information content, see, e.g., [18] and later sections.)

#### 2.1.3. Polarization Selective Interaction

In microscopic samples, anisotropically organized structural elements in a voxel of the sample might be found at nonrandom orientation, i.e., the LD of the latter is not zero. When using a laser beam, which is usually linearly polarized, selective interaction may occur by the preferential/selective excitation of these molecules, which will thus be highlighted in LD_max_ position; in contrast, the same molecules may remain undetected if aligned orthogonally. This artefact can easily be avoided by using a depolarized laser beam, or can be corrected for by changing the polarization plane of the excitation.

Hence, it can be concluded that, via adopting polarization spectroscopy techniques to modern microscopes, they will provide additional, unique, and precise information about the molecular organization of the sample.

### 2.2. DP Spectroscopy and Microscopy

In the following paragraphs, first we define the DP parameters of interest for macroscopic samples, and then show how these techniques can be adapted to microscopy imaging. For the absorption/transmission regime, we confine this treatment to LD and CD, the two most frequently used techniques. The measuring principles of these two techniques are comparable, and in modern spectropolarimeters/dichrographs, the measurements are performed using essentially the same components.

Both LD and CD are defined as the absorption difference (ΔA) between two orthogonally polarized light beams: ΔA = A_1_ − A_2_. In LD, A_1_ and A_2_ refer to absorbances of, e.g., vertically and horizontally linearly polarized light beams, respectively (LD = A_V_ − A_H_). CD = A_1_ − A_2_ = A_L_ − A_R_, where the indices L and R refer to left and right circularly polarized light, respectively. For nonzero LD, a macroscopic sample must be aligned. The orientation angle of the transition dipole with respect to the selected co-ordinate system can be determined using the reduced LD (LD^r^ = LD/A) or the dichroic ratio DR = A_1_/A_2_ (see [19]). CD is defined for isotropic samples.

In modern spectropolarimeters, the measurements are performed with the aid of high-frequency modulation of the polarization state of the measuring light, using a photoelastic modulator (PEM) and a demodulation electronic (usually a lock-in amplifier), which produces ΔA as a function of wavelength; usually A is measured parallel with ΔA. With fluorophores, the differential absorption of the excitation can also be determined via measuring the fluorescence intensity difference between the two emissions which are elicited by orthogonally polarized excitations. These quantities, the fluorescence-detected LD and CD, FDLD and FDCD, also carry information about the (macro)organization of the absorption dipoles, but additional factors, such as energy transfer or excitation energy quenching, may also play roles.

Anisotropic organization of the sample can also be characterized by determining the polarization content of the fluorescence emission. The quantities analogous to LD and CD, respectively, are the (linear) anisotropy (r = (F_90°_ − F_0°_)/F_a_ and (F_+45°_ − F_−45°_)/F_a_) of the fluorescence emission and the circularly polarized luminescence (CPL = (F_L_ − F_R_)/F_a_), where F_a_ is the average fluorescence intensity. Instead of r, often the dichroic ratios are measured (FP = F_90°_/F_0°_ and FP = F_+45°_/F_−45°_). In all cases, nonpolarized excitation must be used to avoid photoselection and contributions by factors determining P, the degree of polarization of fluorescence emission (see below). It must also be noted that, unlike LD or DR, which depend on the non-random orientation of the absorption transition dipoles, r and FP provide information on the anisotropic distribution of the emission dipoles. The same holds true, mutatis mutandis, for the relationship between CD and CPL. (For a technical description, including the techniques of macroscopic alignment of the sample and quantitative analyses, see, e.g., [19].) Anisotropic materials also display linear birefringence (LB), i.e., they possess a refractive index that depends on the polarization and propagation direction of light, which is usually characterized by measuring the phase shift of the linearly polarized measuring light.

The measuring principles and basic techniques outlined above can be applied or adopted for microscopic DP imaging. Note that, when imaging LD or r, the sample does not need to be aligned, especially when it contains a well-discernible reference plane or axis. By this means, microscopic LD or r can be measured even on macroscopically isotropic samples, e.g., on a large vesicle of spherical symmetry [20]. In all cases, full information on the orientation of the transition dipoles can be obtained from two measurements, determining the difference between the transmitted intensities in the horizontal and vertical directions and between the +45° and −45°, respectively. On the other hand, in (FD)CD and CPL measurements, the microscopic alignments may complicate the picture because of the emergence of anisotropic CD (ACD), also known as oriented CD (OCD) [21,22].

With regard to the measuring techniques, the ΔI intensity differences and the I_a_ average intensities can be measured with high precision, pixel by pixel, by using the high-frequency modulation/demodulation circuits—using similar modules as in modern dichrographs, i.e., a PEM and a lock-in amplifier, complemented with some passive polarization optical elements. In simple cases, subtracting the images recorded with the corresponding two orthogonal polarizations may also yield good quality DP images. It is important to emphasize again that fluorescence-detected LD (FDLD) carries information on the alignment of absorbance dipoles, i.e., on the LD of the fluorophore; for complex structures, as pointed out above, additional information can be derived from FDLD measurement, related to different factors affecting the fluorescence emission. The main advantage of FDLD is that, in most common LSMs, it can be measured in confocal regime, whereas LD can only be measured in non-confocal mode (in transmission).

### 2.3. DP Imaging Devices

The first DP microscope was constructed in 1985 by Mickols et al. [23]. It was based on a scanning-stage microscope and was capable of recording LD images in confocal mode. A similar setup was used by Finzi et al. [24] for recording high-resolution LD and CD images of isolated plant thylakoid membranes. The first laser-scanning microscope which was equipped with high-frequency modulation and demodulation circuits was designed for the determination of LB and LD, in non-confocal regime, on ultrathin Langmuir–Blodgett films [25]. The first DP-LSM which was extended to the fluorescence light path, and, thus, in addition to LB and LD, was capable of confocally determining FDLD, r, and P, was designed and constructed in our laboratory [26], and was first used to characterize the anisotropic molecular organization of Alexa-phalloidin-stained actin-based *Drosophila melanogaster* nurse cell samples via mapping of r [27]. In this device, and in similar LSMs of other types and makes, different DP attachments and different configurations were used, but employing the same basic principles and high-frequency modulation and demodulation modules, and passive polarization optical components [10,12]. In all of these DP-LSMs, DP images are recorded parallel with the ‘conventional’ intensity images, and the DP images, after normalization, are saved in the same format as the conventional LSM image. Hence, these images can be overlaid on, and analysed together with the conventional intensity image. While the structural/anisotropy information obtained from DP imaging is comparable to that provided by dichrographs, the data are restricted to discrete wavelengths. On the other hand, DP-LSM measurements provide 2D or 3D microscopic maps of the anisotropy parameters. The use of these devices has been demonstrated on a number of highly organized structures in biological samples and intelligent materials (see [10,12] and references therein).

### 2.4. Information Content of DP Images

The theoretical background of DP imaging has been elaborated by Bustamante and colleagues [13,14,28], showing that different microscopic DP quantities carry important and unique physical information on the anisotropic molecular architecture of the samples. The information content of different quantities which can be determined in DP-LSMs, using linearly polarized light, is summarized in Table 1. Note that the analogous quantities using circularly polarized light, i.e., CPL, CD, and FDCD, as well as circular birefringence (CB), are not listed here—mainly because, in most samples, the magnitudes of these quantities are considerably smaller than the corresponding linear polarization quantities (r, LD, FDLD, and LB, respectively), and also because the determination of these parameters is technically more challenging. Some optical components of CLSMs, especially the dichroic mirrors, may introduce polarization distortions. These should either be avoided by custom compensator plates and/or must be taken into consideration [11].

A recently constructed DP-RCM is capable of confocal DP imaging using fluorescence detection [12]. Since the RCM employs camera-based imaging—instead of using a photomultiplier (PMT) or any other point detection method—the PEM and lock-in modules cannot be used. Instead, FDLD, r, and P can be obtained from sequentially recorded images using rapid polarization-state-adjusting active (liquid-crystal-based) device and passive polarization optical components, either or both of which are inserted in the excitation or the emission beam path. In this case, the stability of the sample is a key factor of the measurements. The effects of minor displacements of the sample can be minimised by applying fast scanning modes (0.5–1 fps for the RCM at 1024 × 1024 or 512 × 512 pixels—newer models are even faster). If necessary, image postprocessing protocols can be applied using cross-correlation-based transformation to match pixel-by-pixel the subsequent images. The two systems, DP-LSM and DP-RCM, can be based on the same platform, as shown in Figure 1.

Similar constructions, widefield, confocal, and multiphoton systems equipped with polarization-sensitive attachments, were used in several studies in basic biology [30,31,32,33,34].

## 3. Applications in Plant Cell Research

Presently, we are aware of only two areas of plant biology applications: (i) chloroplast thylakoid membranes and isolated multilamellar light harvesting complex II (LHCII), and (ii) cell walls and isolated cellulose-based macro-assemblies. A short overview will follow on the background and significance of the anisotropic molecular organization of these structures, and selected data will be shown to demonstrate the use of DP imaging obtained on these units.

### 3.1. Anisotropy and DP Imaging of Granal Thylakoid Membranes

The light reactions of oxygenic photosynthesis occur in, or are closely associated with the thylakoid membranes (TMs). These membranes accommodate the light-harvesting antennas (in plants, LHCII and LHCI), the core antenna complexes, the two photochemical reaction centres (the RC of PSII and the RC of PSI), as well as the cytochrome b_6_f complex, the ATP synthase, and some additional constituents of the photosynthetic electron transport system [35]. The functional activity of this machinery, and thus the efficiency of the light–energy conversion, depends largely on the molecular architecture and macro-organization of the thylakoid membranes.

LD and CD techniques have contributed significantly to our understanding of the organization of the pigment systems in the light-harvesting, core-antenna, and RC complexes and in the TMs. In particular, systematic LD studies, in perfect harmony with structural data, have revealed that, in mature organisms, all pigments in all photosynthetic organisms display nonrandom orientation with respect to each other, to the axes of the protein complexes, and to the membrane normal. The nonrandom orientation of the absorbance and emission dipoles plays important role in the excitation–energy migration processes which feed the RCs. In perfect agreement with this, as has been shown by DP imaging techniques, isolated TMs display strong LD signals when viewed from an angle parallel with the equatorial plane of chloroplasts (edge-aligned TMs); in face-aligned position, much weaker LD signals are detected, which tended to average to LD = 0 for the entire membrane system (Figure 2a,b, respectively). CD images (Figure 2c,d) have revealed that the signals originate from highly organized macro-domains of the TMs—corroborating the notion that the intense, anomalously shaped CD signals of TMs, which are accompanied by differential scattering, are of psi-type origin [24,36] (psi, polymer- or salt-induced). Psi-type aggregates are three-dimensional, chirally organized molecular assemblies, possessing sizes commensurate with the wavelength of the visible light and containing densely packed interacting chromophores [19,37]. The self-assembly, exact physical origin, and physiological roles of these macro-assemblies, which are abundant in biology (e.g., viruses, nuclei, chromosomes, DNA and protein aggregates, multilamellar TMs), are still not understood and, thus, DP imaging techniques might help in elucidating these problems.

These types of studies might also be aided by LB imaging, which can be performed with a configuration of the DP-LSM displayed in Figure 3. The measurements have been performed on isolated, magnetically aligned pea TMs and have shown that edge-aligned granal thylakoid membranes exhibit very strong, but largely heterogeneous, birefringence (Figure 4), which can be correlated with the 3D organization of the TM system [38,39,40]. The much weaker LB of face-aligned chloroplasts is explained by the symmetry of the membrane system, and is in agreement with LD data. The strong LB of TMs, as well as small subchloroplast membrane particles and lamellar aggregates of LHCII, have been shown to facilitate their optical trapping and alignment and micromanipulation using a linearly polarized Gaussian-laser-beam trap [40].

### 3.2. The Significance of Confocal DP Imaging for Plant Cell Wall Research

The biomass stored in plant cell walls is one of the most abundant energy resources in the biosphere. However, presently, it appears to be extremely difficult to convert the stored energy of lignocellulosic biomass into biofuel. This difficulty, to a large extent, arises from the intrinsic resistance of cell walls to decomposition into sugars. This underlines the importance of research on the architecture of cell walls and, particularly, their anisotropic molecular organization, the significance of which has been recognized for many years (cf. [41] and references therein).

The cell wall is a plant cell’s compartment lying outside the plasmalemma. Its biological function is to provide mechanical support and protect the cell from stress. The composition of cell walls varies between cell types. All cell walls contain two layers, the middle lamella and the primary cell wall, and many cells produce a secondary wall. Wood cell walls are built of several layers containing an ordered array of cellulose fibrils organized in microfibrils and embedded in a matrix of polysaccharides (pectin, hemicellulose) and lignin [42,43]. The distribution and orientation of cellulose microfibrils is determined by both genetic and abiotic factors. Genetic factors comprise cell wall layers (primary wall, S1, S2, S3 layers of secondary cell wall), position (radial or tangential cell wall), plant age (juvenile or mature wood), and season of maturation within the growth ring (early and late wood), whereas abiotic factors comprise mechanical stress caused by wind or stem lean [44,45]. This is why one of the most frequently measured ultrastructural variables in the wood cell wall is microfibrillar angle (the angle between the tracheid axis and the cellulose microfibrils as they coil around the cell) [46]. Structural organization of the cell wall and related polymers is important for both mechanical properties of the plants and chemical reactions occurring in the cell walls, especially in response to stress.

#### 3.2.1. Convallaria Majalis Root Cell Walls

Following the acquisition scheme shown in Figure 5, both transmission and fluorescence lights were detected and analysed by the phase sensitive lock-in amplifier. As shown in Figure 6, thin sections of the rhizome of *Convallaria majalis* stained with acridin orange [47] exhibit nonzero LD at 488 nm. However, this signal was weak and smeared out (Figure 6a,b shows the transmission), probably because of the layer-by-layer disorder in the sample—lowering the observed averaged anisotropy. It has been reported that adjoining cell walls might contain microfibrils with different (or even opposite) preferential orientation, which, thus, can lead to a decrease in birefringence retardation [48]. In thick and dense preparations, a further complication might arise from artefacts due to light scattering. These limitations of non-confocal LD imaging can be overcome by applying confocal imaging [49].

As shown in Figure 6c, the confocal image of fluorescence removes the blurriness and gives clear contours. The FDLD images (Figure 6d) clearly show that the preferential orientation of the cellulose fibrils, visualized with acridin orange, closely follows the orientation of the cell wall. It was positive for vertically oriented sections, negative for horizontal orientations, and vanished for diagonal sections. It has also been confirmed that, upon rotating the sample by 90° and 45°, the anisotropy changed sign (90°) and the “diagonal” dichroic features became highlighted (45°). This is fully in line with the expectations and shows that FDLD was free of artefacts. Further corroboration of these data were obtained by r images, i.e., by mapping the anisotropic distribution of the emission dipoles. These data are in accordance with the polarization microscopy data of Verbelen and colleagues, who revealed the net orientation of cellulose fibrils in the outer epidermal layer, using Congo Red and polarization confocal microscopy [50].

By taking advantage of confocal imaging, FDLD images can be successfully used for optical sectioning and the reconstruction of anisotropy in 3D (Figure 6e). As shown in the gallery, the anisotropy is retained in the z direction but the layers do not overlap perfectly, due also to shape variations. These variations, while explaining the blurred images of the (non-confocal) LD signal, underline the significance of confocal FDLD (or r) imaging technique(s) (cf. Figure 2b).

As demonstrated by this example, anisotropy imaging in DP-LSM yields high precision data, which are suitable for quantitative data analyses and model calculations, and thus can be used to determine the orientation angle of the absorption (FDLD) or emission (r) dipoles of the intercalated dye molecules with respect to the axes fixed to the co-ordinate system attached to the sample.

#### 3.2.2. Comparison of Cell Wall Structure in Stems of Different Plant Species

Taking advantage of the high precision, quantitative FDLD values obtained from DP-LSM imaging (Figure 7), anisotropic features of different plant cell walls (hardwood: maple—*Acer platanoides* L., softwood: spruce—*Picea omorika* (Pančic) *Purkyne* and maize—*Zea mays* L.) can be compared [51]. This was performed via analysing the distribution of FDLD values indicating the degree of order in the cell walls. It was shown that all three species have similar cellulose orientations in the cell walls (Figure 8). However, the narrowest distribution was obtained for maize, indicating the simplest structure and more regular packing of cellulose molecules in maize than in the wood species. This might be of significance in relation to its digestion and decomposition into smaller polysaccharides.

#### 3.2.3. Parenchyma Cell Wall Structure in Liana Plants

The aim of this study was to determine changes in stem parenchyma cell walls related to the twinning mechanism of liana plants [52]. During the analysis, parenchyma cell walls parallel to the Y-axis were chosen. FDLD microscopy (Figure 7) empowered screening and evaluation of the main differences in cellulose fibrils alignment compared to one another in the X–Y plane (“cellulose fibrils order”) in parenchyma cell walls of straight (1st–4th) and twisted (5th) internodes of *Dioscorea balcanica*. The FDLD distributions and corresponding FDLD values (Figure 9), as well as one-way ANOVA of the maximum positions of Gaussian curves between internodes, showed no significant distinction in “cellulose fibrils order” in parenchyma cell walls related to mechanical strain in twining stem, i.e., between straight (1st–4th) and twisted (5th) internodes of *D. balcanica*.

#### 3.2.4. Cell Wall Response to Mechanical Stress

As a further use of quantitative FDLD imaging (Figure 7), the response to mechanical stress was analysed on wood cell walls of juvenile Serbian spruce [53]. FDLD microscopy was used to compare the distribution and alignment of cellulose fibrils in cell walls of compression (CW) and normal wood (NW), on stem cross-sections of juvenile *Picea omorika* trees. These investigations revealed significant differences in cellulose fibril order between NW and CW, and also between radial and tangential walls (Figure 10). As above, blue represents dipoles (strained cellulose fibrils) which absorb light polarized predominantly parallel with the X-axis, and yellow marks fibrils which tend to be oriented parallel with the Y-axis; grey tones represent fibrils oriented at about 45°. The analysed stem cross-sections were oriented in such a way that each observed tracheid tangential wall is oriented parallel to the X-axis, whereas the radial wall is oriented parallel to the Y-axis, and the tracheid axis corresponds to the Z-axis of this co-ordinate system (Figure 10q,r).

The FDLD images have shown that, in tangential walls of NW, cellulose fibrils are oriented mostly parallel to the X-axis (blue coloured) in all cell wall layers (Figure 10a). In contrast, in CW samples, the cellulose fibrils are oriented parallel to the X-axis only in the outer cell wall layers, whereas, in the innermost part of the S2 layer (Figure 10e,i,m), the fibrils are more disordered. Similar but much less pronounced differences were seen in radial walls, between NW and severe CW for the regions containing fibrils parallel to the Y-axis (yellow-coloured) (Figure 10a,i,m). This is in accordance with the main severe CW characteristics: reduced level of cellulose deposition in the S2 layer, and the absence of an S3 layer [44]. In the sample of mild CW (Figure 10e), differences in cellulose fibril alignment/order compared with NW are slightly less expressed in both radial and tangential walls.

Using ImageJ [54] macros on DP images, the differences in FDLD distribution and, thus, relative distribution and order of cellulose fibrils within tangential and radial walls in NW and CW were evaluated and quantified (Figure 10s,t). It was found that, in NW, FDLD distributions were narrow in both tangential and radial walls, with maxima (Figure 10c,d) positioned in blue (fibrils predominantly more parallel to X-axis) and yellow (fibrils predominantly more parallel to Y-axis), respectively. In CW, there was a widening of FDLD distribution and shift of pixels to grey. It could thus be concluded that the number of disorientated fibrils was increasing in both the tangential (Figure 10g,k,o) and the radial walls (Figure 10h,l,p) compared to NW (Figure 10c,d). This indicated that order/alignment of cellulose fibrils in CW decreases compared to NW.

## 4. Conclusions and Perspectives

The major aim of this review paper is to call the attention of researchers in the field of plant cell imaging to differential polarization imaging techniques, which have been demonstrated to provide valuable, unique information about the anisotropic molecular organization of chloroplast thylakoid membranes and cell walls. Microscopic DP imaging systems, especially when combined with modern laser-scanning microscopy techniques, as well as with spectroscopic tools, hold the promise of exploring further details of the molecular organization of plant cells and their constituents. This expectation is based on the notion that full, or at least a more comprehensive, understanding of light–matter interactions requires the use of polarization spectroscopy methods. Multifaceted interactions between polarized light and highly organized biological samples evidently contain a wealth of information which cannot be obtained with nonpolarized light. Thus, micro-spectropolarimetry may open up new vistas to revealing the self-assembly, molecular organization, and structural dynamics of a range of plant materials and plant-mimicking self-assembly materials [55]. The following examples illustrate the potential use of DP imaging. In photosynthetic tissues, the remodelling of chloroplasts and the thylakoid membranes can be monitored by FDCD, CD spectroscopy, which has been thoroughly demonstrated to be a sensitive indicator of membrane reorganizations [56]. FDLD might be a suitable noninvasive tool to monitor chloroplast photorelocation movements, which, in terrestrial angiosperms, have been shown to be governed by blue-light, perceived by phototropins, and actin filaments [57,58]. DP microscopy may aid in obtaining refined structural information on the organization of plant cytoskeleton [59]; DP-LSM has been shown to reveal protein-mutation-induced reorganization of the actin-based nurse cell canals of *Drosophila melanogaster* [27]. Microviscosity of the cytoplasm might possibly be determined via P imaging using a properly selected dye molecule; the same holds true for lipophylic fluorophores and lipid membranes (e.g., of mitochondria or the Golgi apparatus). Although it may be challenging, nuclei might be interesting targets to test with CPL and FDCD. Micromanipulation of birefringent cell components, using linearly polarized light [40,60], might be useful in biotechnological applications.

Further developments of the DP imaging techniques can be envisioned to enhance the available optical resolution and to extend the measurable polarization-dependent parameters, in the best-case scenario including all the Mueller matrix elements. The RCM unit provides the possibility (the place and the electrical connections included) to apply a more complex polarization state generator and polarization state analyser instead of the currently installed liquid-crystal-based modulator.

To sum up, Table 2 shows a list of recent applications of DP-LSM and DP-RCM, providing an overview of the DP quantities determined and a selection of investigated biological and biomimetic materials.

## Figures and Tables

**Figure 1 ijms-22-07661-f001:**
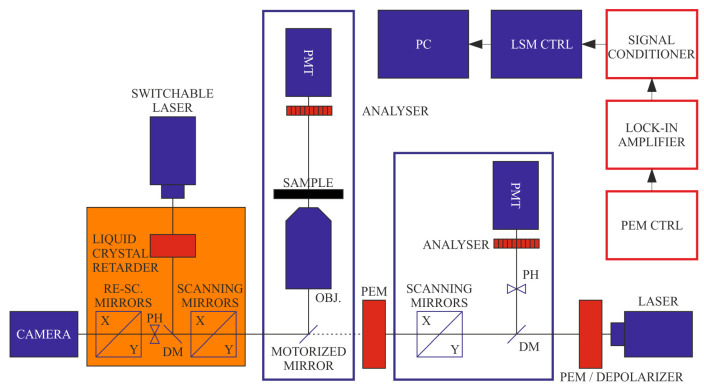
Schematics of the combined set-up of DP-LSM and DP-RCM, using a common platform of an inverted fluorescence microscope. Units of the ‘conventional’ LSM and RCM are displayed in blue and orange colours, respectively; red boxes are the components of the DP attachments.

**Figure 2 ijms-22-07661-f002:**
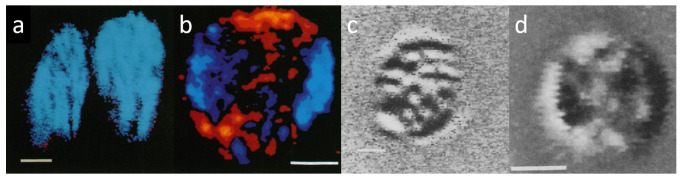
Confocal LD and CD images ((**a**–**d**) respectively) of isolated magnetically oriented spinach TMs trapped in edge-aligned (**a**,**c**) and face-aligned (**b**,**d**) positions. Blue and red false colours stand, respectively, for positive and negative LD with respect to the membrane plane (**a**,**b**); black and white domains (**c**,**d**) indicate local CD signals of opposite signs. Bar is 2 µm. (Composed from parts of figures published in [24]).

**Figure 3 ijms-22-07661-f003:**
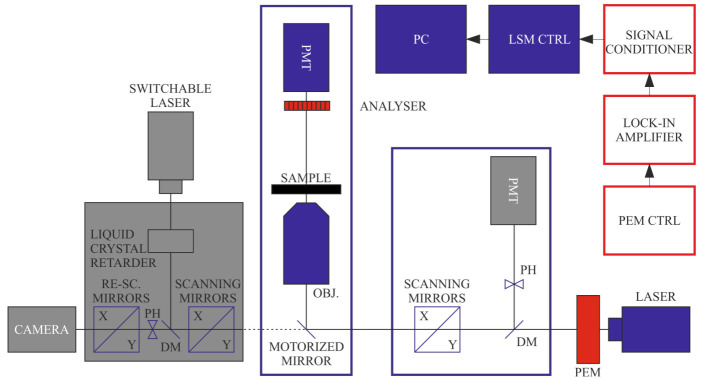
Schematics of LB imaging using the DP-LSM setup, using the transmission detector, in this case, in non-confocal regime, and an analyser applied in front of the detector. (Components inactive in this configuration are displayed in grey).

**Figure 4 ijms-22-07661-f004:**
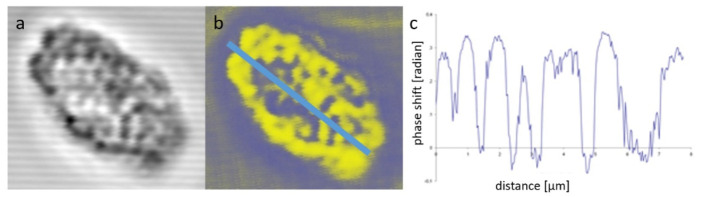
Non-confocal transmission (**a**) and LB (**b**) images of an isolated, edge-aligned TM system of a granal (pea) chloroplast; and variations of the phase shift (LB) (**c**) along the axis of the TM system. (Composed from parts of a figure published in [40], with permission from Springer Nature).

**Figure 5 ijms-22-07661-f005:**
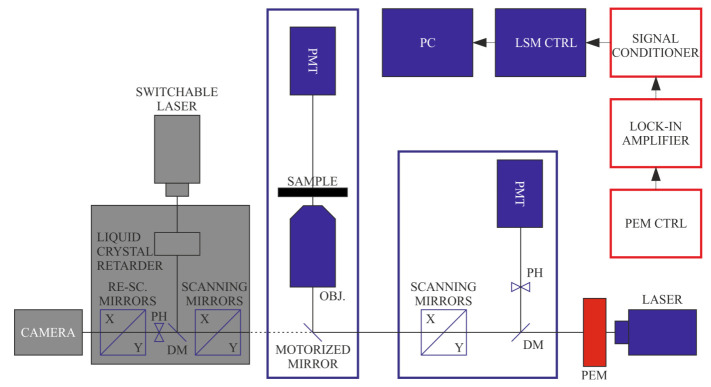
Configuration of DP-LSM for (non-confocal) LD and (confocal) FDLD imaging. Independent acquisitions of LD and FDLD were performed by using transmission and fluorescence detectors, respectively. (Components inactive in this configuration are displayed in grey).

**Figure 6 ijms-22-07661-f006:**
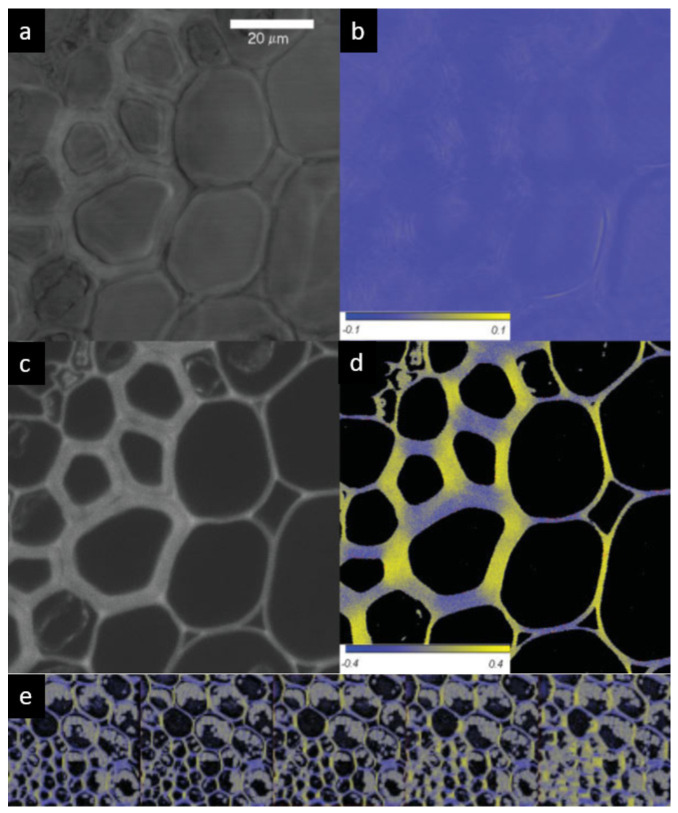
Non-confocal transmission (**a**) and orientation factor (S = LD/3A) (**b**) images, and confocal fluorescence intensity (**c**) and FDLD (**d**) images of thin sections of the root of *Convallaria majalis* stained with acridin orange. False colours and scales in (**b**,**d**) indicate the sign and magnitude of the dichroism. (**e**) Gallery of 3D reconstructed FDLD images. Bar is 20 µm. (Composed from figures published in [49], with permission from John Wiley and Sons).

**Figure 7 ijms-22-07661-f007:**
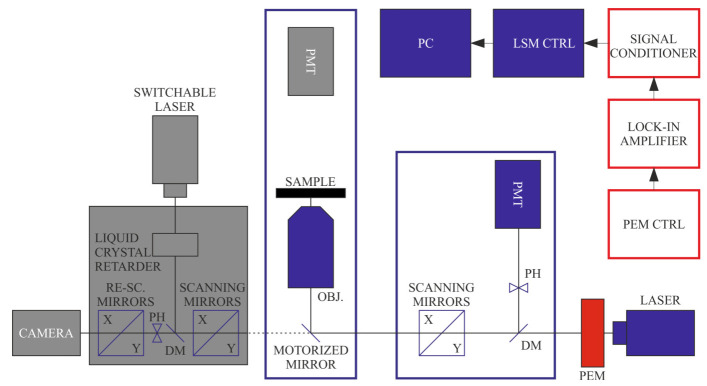
For cell wall structure investigations, we have applied the confocal FDLD imaging with polarization-modulated excitation and using fluorescence detection. (Components inactive in this configuration are displayed in grey).

**Figure 8 ijms-22-07661-f008:**
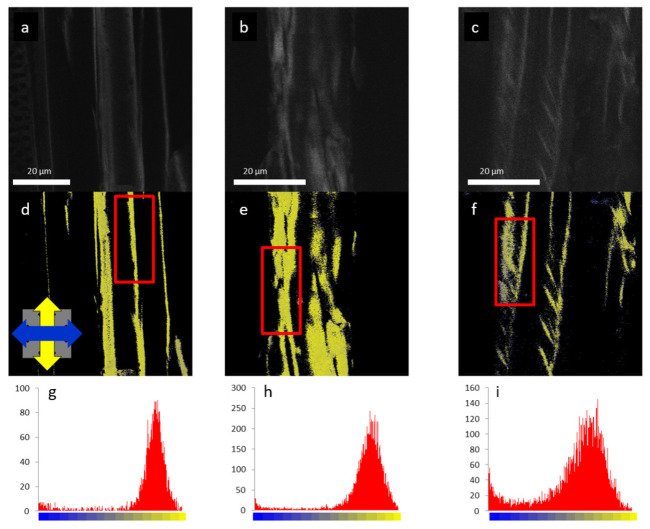
Confocal fluorescence intensity (upper panels) and FDLD images (middle panels) of isolated cell walls of maize (**a**,**d**,**g**), maple (**b**,**e**,**h**), and spruce (**c**,**f**,**i**) stained with Congo red; and FDLD distributions measured on the framed regions (lower panel). The blue and yellow regions in FDLD images are for the horizontal and the vertical dipole orientations, respectively. (Reproduced in colour from [51], with permission from Springer Nature).

**Figure 9 ijms-22-07661-f009:**
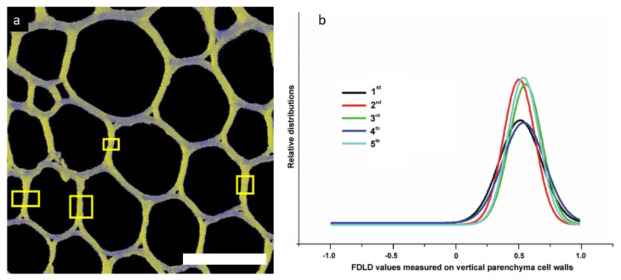
(**a**) Example of DP-LSM image (3rd internode, straight), with marked yellow rectangles presenting parts of cell walls used for FDLD analyses. Bar is 20 µm. (**b**) Gaussian fits of FDLD distributions in *D. balcanica* parenchyma cell walls of straight (1st–4th) and twisted (5th) internodes. (Composed from figures published in [52], with permission from Springer Nature).

**Figure 10 ijms-22-07661-f010:**
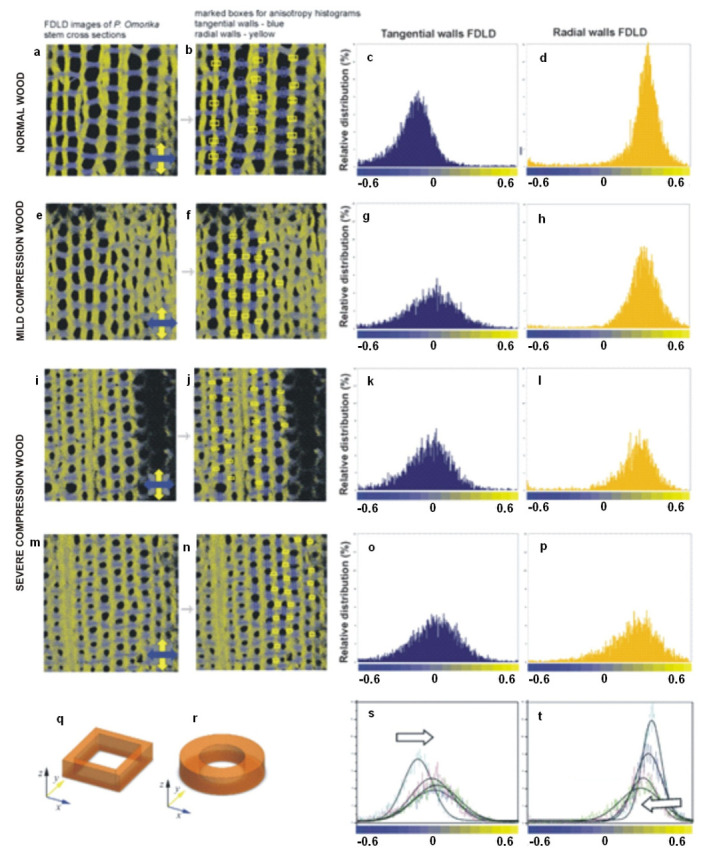
FDLD images of *Picea omorika* stem sections and corresponding anisotropy histograms. (**a**) Normal wood (NW); (**e**) mild CW; (**i**,**m**) severe CW; (**b**,**f**,**j**,**n**) pixel values were collected in the marked areas, in tangential walls (blue boxes) and in radial walls (yellow boxes), and used to obtain anisotropy distributions; (**c**,**g**,**k**,**o**) tangential walls FDLD distributions (**d**,**h**,**l**,**p**), radial walls FDLD distributions; (**s**,**t**) overlaid distributions with black lines representing corresponding Gaussian fits (white arrow represents gradual shifts toward grey—increasing number of disorientated fibrils); (**q**,**r**) schemes of NW and CW tracheids, respectively. Excitation at 488 nm, emission above 560 nm. The blue and yellow colours of the co-ordinate axes correspond to the orientation of tangential and radial walls in the images, respectively. (Published in [53], with permission from Cambridge University Press).

**Table 1 ijms-22-07661-t001:** Features of the main DP quantities which can be imaged in a confocal LSM via using linearly polarized light. For LD, FDLD, and r, I_1_ and I_2_ are the measured transmitted (LD) or fluorescence (FDLD, r) intensities with the requested polarization states with respect to a plane or axis, which is usually defined by the sample; for P, they are related to the plane of the polarization of the excitation beam; I_0_ is the incident light intensity; A_x_ is absorption, where x (=1, 2) refers to the appropriate polarization state. ΔI and I_a_ are the measured differential and average intensities, respectively. The term, “Modulated” refers to the modulation of the polarization state, between orthogonal states, of the excitation beam or the detected fluorescence emission. LB imaging requires the determination of the phase shift of a linearly polarized beam along a predetermined axis, see [29] and references therein.

DP Quantity	Laser Beam Polarization	Detection	Definition	Information Content
LD	Modulated	Nonpolarized. Transmission: non-confocal	Linear dichroism A_1_ − A_2_ ≈ (I_2_ − I_1_)/2.3 I0	Anisotropic organization of molecular macro-assemblies (absorption dipoles)
FDLD	Modulated	Nonpolarized. Fluorescence: confocal	Fluorescence-detected linear dichroism I_1_(A_‖_) − I_2_(A_⟂_) = ΔI	LD of fluorophores (absorption dipoles)
r	Unpolarized	Modulated. Fluorescence: confocal	Anisotropy of the emission dipoles: (I_1_ − I_2_)/I_a_	Anisotropic organization of molecular macro-assemblies (emitting dipoles)
P	Linearly polarized	Modulated. Fluorescence: confocal	Degree of polarization of the fluorescence emission: (I_1_ − I_2_)/(I_1_ + I_2_)	Microviscosity, fluorescence lifetime, energy transfer
LB	Modulated	Polarized	Phase-shift determination	Anisotropic ultrastructure or texture

**Table 2 ijms-22-07661-t002:** An overview of recent DP-LSM and DP-RCM applications on different materials, and related publications.

DP Quantity	Materials/Objects	Plant Biology	Publications
LD	*Convallaria majalis* root cell wall	yes	[49]
FDLD	*Acer platanoides* L., *Picea omorika, Zea mays* L.	yes	[51]
	*Dioscorea balcanica*	yes	[52]
	Isolated human amyloid	no	[61]
	Synthetic porphyrin microrods (Zn-C2)	no	[55]
r	Synthetic porphyrin microrods (Zn-C2)	no	[55]
	*Drosophila melanogaster* nurse cells ring canals	no	[27]
P	JY human B lymphoblasts	no	[20]
LB	Chloroplasts isolated from *Pisum sativum*	yes	[29]
FDLD (on DP-RCM)	*Convallaria majalis* root cell wall	yes	[12]

## Data Availability

Not applicable.

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
