# Peer review of "Differential Polarization Imaging of Plant Cells. Mapping the Anisotropy of Cell Walls and Chloroplasts"

_ijms, 2021, doi:10.3390/ijms22147661_

Round 1

Reviewer 1 Report

Thank you for the possibility to review this manuscript, I read it with a great interest. The manuscript provides comprehensive information on the differential polarization imaging methodology and areas of its application in studies on plant cell structure and functioning. Authors provide also background knowledge on the technical and analytical basis of DP imaging. In my opinion manuscript covers the important topic in plant cell biology and gathers important information about the technology for scientific community.

Prior to publication I would recommend following:

-presentation of the new and perspective study fields in which DP imaging techniques could be applied in a form of table - it is very important paragraph since it shows future directions for other researchers to implement the technology and it is worth to be highlighted and eye-catching (conclusions lines:457 and subsequent)

- provide an information on the drawbacks and challenges in application of DP imaging techniques in biological research

Author Response

First of all, we would like to express our gratitude to Reviewer 1 for reading and commenting on our manuscript. Following his/her suggestions, we have modified the manuscript. You will find our answers listed point-by-point below.

Point 1: presentation of the new and perspective study fields in which DP imaging techniques could be applied in a form of table – it is very important paragraph since it shows future directions for other researchers to implement the technology and it is worth to be highlighted and eye-catching (conclusions lines:457 and subsequent)

Response 1:

We have constructed a summary in table format to make the overview of the recent DP-applications easier to identify.

Point 2: provide an information on the drawbacks and challenges in application of DP imaging techniques in biological research

Response 2:

Using the different confocal-microscopy based DP imaging technologies we have to face different challenges. Some optical components of CLSMs, especially the dichroic mirrors, may introduce polarization distortions. These should either be avoided, by carefully selecting the components, and/or must be taken into consideration (Image post-processing). In case of sequential imaging (collecting multiple parameters from the same area, e.g. r and FDLD; based on two scans, using the camera based DP-RCM, or LB using PEM and sequentially recorded images) the stability of the sample is a key factor of the measurements. The effects of minor displacements of the sample can be minimised by applying fast scanning modes (0.5-1 fps for the RCM at 1024x1024 or 512x512 pixels – newer models are even faster). If necessary, image post-processing protocols can be applied using cross-correlation based transformation to match the subsequent images pixel-by-pixel.

This information is added to the corresponding chapters of the manuscript.

Thanks again for the helpful comments!

Reviewer 2 Report

I believe it a small, succinct, and nicely written review on a rather specific subject of interest. It is a nicely conducted review and I see no mistakes.

What I would expect from such a review is more images. I think it would be helpful to add more images indicating specific techniques that are reviewed.

What is more, English needs a little polishing. A native reader should conduct some corrections.

Therefore a revision is required.

Author Response

We would like to thank Reviewer 2 for reading and commenting on our manuscript. Following his/her suggestions, we have revised our paper. You will find our answers listed below.

Point 1: What I would expect from such a review is more images. I think it would be helpful to add more images indicating specific techniques that are reviewed.

Response 1:

In the revised manuscript, further schemes were added to identify the specific techniques used in the DP-measurements shown.

Point 2: What is more, English needs a little polishing. A native reader should conduct some corrections.

Response 2:

Professional English language assistance has been engaged to improve the text.

Thanks again for the comments of Reviewer 2!